# UAV-Based Hyperspectral Imaging for River Algae Pigment Estimation

Riley D. Logan [1], Madison A. Torrey [2], Rafael Feijó-Lima [3], Benjamin P. Colman [3], H. Maurice Valett [4] and Joseph A. Shaw [1,*]

[1] Optical Technology Center, Electrical and Computer Engineering Department, Montana State University, Bozeman, MT 59717, USA; riley.logan@montana.edu

[2] Civil Engineering Department, Montana State University, Bozeman, MT 59717, USA; madison.torrey@montana.edu

[3] Department of Ecosystem and Conservation Sciences, University of Montana, Missoula, MT 59812, USA; rafael.feijo@mso.umt.edu (R.F.-L.); ben.colman@mso.umt.edu (B.P.C.)

[4] Division of Biological Sciences, University of Montana, Missoula, MT 59812, USA; maury.valett@umontana.edu

[*] Correspondence: joseph.shaw@montana.edu

**Abstract:** Harmful and nuisance algal blooms are becoming a greater concern to public health, riverine ecosystems, and recreational uses of inland waterways. Algal bloom proliferation has increased in the Upper Clark Fork River due to a combination of warming water temperatures, naturally high phosphorus levels, and an influx of nitrogen from various sources. To improve understanding of bloom dynamics and how they affect water quality, often measured as algal biomass measured through pigment standing crops, a UAV-based hyperspectral imaging system was deployed to monitor several locations along the Upper Clark Fork River in western Montana. Image data were collected across the spectral range of 400–1000 nm with 2.1 nm spectral resolution during two field sampling campaigns in 2021. Included are methods to estimate chl *a* and phycocyanin standing crops using regression analysis of salient wavelength bands, before and after separating the pigments according to their growth form. Estimates of chl *a* and phycocyanin standing crops generated through a linear regression analysis are compared to in situ data, resulting in a maximum $R^2$ of 0.96 for estimating fila/epip chl-a and 0.94 when estimating epiphytic phycocyanin. Estimates of pigment standing crops from total abundance, epiphytic, and the sum of filamentous and epiphytic sources are also included, resulting in a promising method for remotely estimating algal standing crops. This method addresses the shortcomings of current monitoring techniques, which are limited in spatial and temporal scale, by proposing a method for rapid collection of high-spatial-resolution pigment abundance estimates.

**Keywords:** remote sensing; algal pigment estimation; linear regression; imaging systems; hyperspectral imaging; drones; UAVs; river algae; algal blooms; water quality; inland waters

## 1. Introduction

### 1.1. Background

In 1983, the largest collection of US Environmental Protection Agency Superfund sites was established to include the Upper Clark Fork River (UCFR), a snowmelt-fed river located in western Montana, USA. The river's headwaters were the locus of over 100 years of mining and smelting activity, the byproducts of which were repeatedly deposited along the river and its adjacent floodplain via recurrent flooding. Along with heavy metal contamination, the UCFR suffers from anthropogenic nitrogen enrichment that, coupled with naturally high levels of phosphorous availability and insolation, leads to annual nuisance blooms of filamentous algae (*Cladophora glomerata*) during periods of enhanced net primary productivity [1–3]. Furthermore, as the growing season progresses, *Cladophora*

is replaced by bluegreen algal growth (Valett et al., in review). As these dense mats of algae grow, they can disrupt dissolved oxygen and pH [4,5], reduce benthic biodiversity [6], deteriorate water quality [4,7–9], and decrease recreational opportunities [4,7,9]. To understand the ecological status, composition, and health of water bodies (i.e., water quality) in which riverine algal blooms (RABs) occur, the standing crops of various algal pigments are commonly measured, rendering information on the extent and duration of the bloom. Monitoring RAB development is particularly important in Montana, which was one of the first states to adopt water quality standards based in part on numeric algal biomass, measured as standing crops of benthic algal chl *a* [10].

Photosynthesis in algae is supported by three major classes of pigments: chlorophyll, carotenoids, and biliproteins. Among the major classes of pigments, chlorophyll *a* (chl *a*) is commonly used to assess bloom characteristics for an array of algal types, while phycocyanin is used for cyanobacteria [11,12]. Methods for measurement generally require gathering organic material in situ and processing in the laboratory. While biomass abundance can be determined by burning the sample to obtain measures of ash-free dry mass (AFDM), pigment content is determined by extraction using various solvents [13]. Current methods of assessing algal abundance via pigment and organic matter standing crops include several shortcomings: time-intensive data collection, limited spatial distribution, and often long delays between sample collection and results. In response, remote sensing has been introduced as a promising method to overcome many of the shortcomings of these traditional techniques [14].

### 1.2. Existing Methods

Remotely sensed imagery, from either satellite or airborne platforms, allow for rapid data collection over large areas with relatively short data processing times [15]. Satellite remote sensing has been used extensively to monitor water quality and assess algal bloom dynamics in oceans [16–22] and to study algal pigments in large lakes [21–24]. However, satellite sensing of rivers and small lakes is prevented or impaired by disadvantages that include coarse spatial resolution, long revisit times, and obscuration by clouds. Unoccupied aerial vehicles (UAVs) have provided a means of capturing high-spatial-resolution image data while also maintaining precise control over the sampling location and time, with improved repeatability and data collection times [25].

UAV-based remote sensing systems have gained popularity in recent years for monitoring algal pigment concentrations in oceans [26], lakes and reservoirs [27–29], and rivers [30–32]. In smaller running water systems, UAV-based multispectral imagers have been used to estimate pigment concentrations using spectral indices such as the normalized difference vegetation index (NDVI), which was shown to be strongly correlated with chl *a* concentration [31]. Remote sensing systems have also been successfully deployed along larger water systems, such as the UCFR, where a small UAV-based color imager with red, green, and blue (RGB) channels was used to assess percent algal cover [30]. UAV-based multispectral imagers also have been used with machine learning techniques to measure chl *a* concentration and total suspended solids in reservoirs and artificial lakes [32].

Spectral indices are commonly used to estimate biological markers in vegetation, ranging from assessing forest health and stress [33] to estimating leaf chlorophyll [34]. These indices vary from simple ratios of two spectral bands to more complex formulations involving several wavelengths. Though spectral indices have been developed for estimating chl *a* concentration in coastal waters [20], inland lakes [35], and streams [31], they have been developed to estimate concentrations within a water body (i.e., per unit volume of water) rather than per unit area. In addition, several established spectral indices have been shown to have decreased performance when generalized across systems with differing water properties [36]. Furthermore, no known studies have assessed water quality through estimation of algal standing crops (i.e., the mass of algal pigment in a given area) using UAV-based hyperspectral imaging systems over shallow rivers.

*1.3. Ecological Context*

While many rivers and streams are heavily canopied and their food webs are subsidized by inputs of terrestrial organic matter [37–42], others, such as streams in semi-arid systems [37] or grasslands [43], and mid-order rivers with open, well-lit channels [44], rely on autochthonous organic matter production and processing [45–47]. In these types of streams, autotrophic organisms (i.e., benthos) typically colonize and grow on the stream bottom and proliferate when warm temperatures, ample insolation, and abundant nutrients support growth. The proliferation of benthic algae influences rates of primary production and respiration, trophic relationships among consumers such as insects and fish, and alters water quality. As systems dominated by the flow of water, measures of ecosystem metabolism [48,49] and nutrient cycling in streams and rivers reflect transport and processing that typically occur over kilometers of river distance upstream from points of assessment. Accordingly, the ability to assess the spatial and temporal distributions of benthic algae is critical for understanding in-stream influences over riverine process. Current methods for quantifying algal standing crops are constrained to scales smaller ($cm^2$–$m^2$) than are needed to couple structure to functional attributes like metabolism or nutrient uptake that extend over larger scales (i.e., 100–1000 m) reflecting the influences of flow. Such constraint makes typical measures susceptible to site-specific conditions (i.e., local variation in growth conditions), rendering extrapolation to larger scales inappropriate.

*1.4. Objectives*

Here, this methodological knowledge gap is addressed by presenting an approach for quantifying algal abundance from image data collected by a UAV-based hyperspectral imaging system to determine standing crops of chl *a* and phycocyanin. Data were collected from the UCFR during the annual algal bloom dominated by the nuisance filamentous green algae, *Cladophora*, but include colonists that live on algal filaments (i.e., epiphytic forms) and more adenate forms associated with rock surfaces (i.e., epilithic algae), and bluegreen cyanobacteria that contribute to periphyton composition as the bloom progresses (Valett et al., in revision). Concurrently, ground-truth data were collected from designated plots to superimpose in situ sampling locations and corresponding image pixels. Using these data sets, the following research objectives were pursued:

1. present a framework for collecting high-spatial and -spectral image data using a UAV-based hyperspectral imaging system along a clear and shallow river;
2. determine optimal linear relationships between spectral band ratios (i.e., the ratio of reflectance values at two wavelengths) and the algal pigments chl *a* and phycocyanin present among filamentous, epiphytic, and epilithic forms of benthic algae;
3. compare the performance of the derived optimal linear relationships to select existing spectral indices developed for monitoring chl *a*.

This paper expands on a conference paper that provided a preliminary summary of this work [50], by including expanded discussions of the ecology of the Upper Clark Fork River; algae phenology; ground-truth data collection; imager calibration; the theoretical basis for the employed method; algorithmic development; results of ground-truth data collection; expanded modeling by separating by algal growth form, including another sampling location, and estimating phycocyanin abundance; an uncertainty analysis to aid in assessments of generalizability; and performance comparisons with several established spectral band ratios.

## 2. Materials and Methods

*2.1. Study Sites*

The UCFR is a clear and shallow cobble-bed river that forms at the confluence of Warm Springs Creek and Silver Bow Creek, with a long history of eutrophication and heavy algal growth [1–3]. Data were collected at two sites on the main stem of the river (Figure 1). The field sites, Gold Creek (46.59°N, 112.93°W) and Bear Gulch (46.70°N, 113.43°W),

feature channels of comparable width (30 and 28 m) and depth (35 cm) during the time of field sampling and both have been shown to contain elevated nutrients, likely from wastewater and other anthropogenic activities in the area [51]. Coupled to little shading from the surrounding riparian canopy, such enrichment creates conditions conducive to RAB proliferation [2].

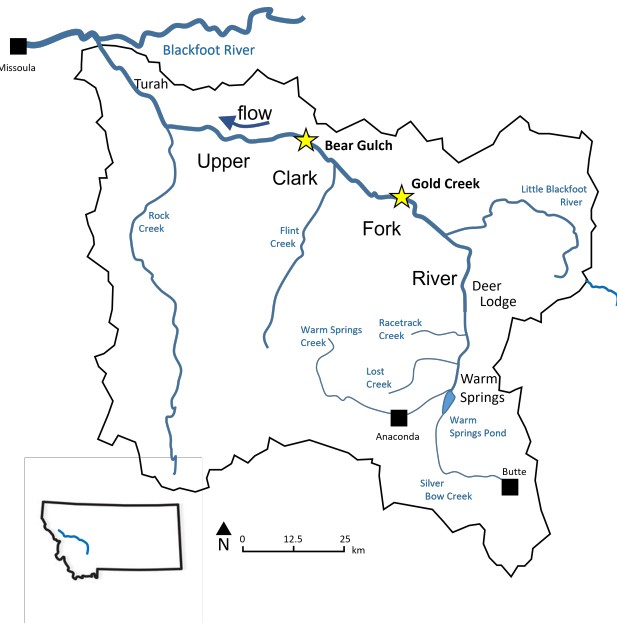

**Figure 1.** The Upper Clark Fork River originates at Warms Spring and flows approximately 200 km downstream to its confluence with the Blackfoot River. Stars indicate sample sites located on the main stem at Gold Creek and Bear Gulch.

### 2.2. RABs and Algal Phenology

Growth of *Cladophora* in the UCFR is controlled by many factors, but is typically initiated after spring runoff events in May or June [52]. Peak runoff scours benthic organic matter through shear forces created by elevated flow. Algal biomass then accrues as flow declines (Valett et al., in press) [53], temperatures increase, and *Cladophora* standing crops become visually evident [54]. The UCFR typically approaches base flow by early July, after which *Cladophora* blooms reach peak biomass [53]. As blooms progress, diatoms begin to accumulate on *Cladophora* filaments, as seen in other river systems experiencing RABs dominated by *Cladophora* [55,56]. Self-shading, grazing pressures, and thermal stress can lead to rapid extirpation and *Cladophora* decline. During this process, *Cladophora* blooms experience a shift in their surface appearance, transitioning from light green (early phases) to yellow (later stages with epiphytic coloration) to a darker brown (heavily colonized or senescent) [57]. As *Cladophora* blooms senesce in late summer and early fall, they give way to mats of bluegreen algae (Valett et al., in press), which uniquely employ the accessory pigment phycocyanin. The progression of RABs in the UCFR is spatially variable with stages of the bloom differing among locations. The visible change in *Cladophora* blooms, including accrual of epiphytic algae and change in physiological status, along with the shift to bluegreen bacteria and its exclusive pigment, phycocyanin, form the basis for remotely sensing algal standing crops of different composition and character.

### 2.3. Imaging System
#### 2.3.1. Imager

Hyperspectral images were gathered using a Pika L hyperspectral imaging system (Resonon Inc., Bozeman, MT, USA), with a spectral range of approximately 387–1023 nm and nominal band spectral resolution of 2.1 nm, resulting in 300 spectral channels. During data collection, the Pika L was fitted with a 17 mm objective lens, resulting in a 17.6° across-

track full-angle field of view. This pushbroom imager was combined with an airborne imaging system that included a system control unit with onboard storage, GPS and inertial measurement unit, and downwelling irradiance sensor, forming the payload for the UAV.

### 2.3.2. UAV

The airborne hyperspectral system described above was mounted to a DJI Matrice 600 Pro hexacopter (DJI, Shenzhen, China) via a DJI Ronin-MX gimbal. The Matrice 600 Pro measures $1668 \times 1518 \times 727$ mm, with all frame arms, propellers, and GPS antennae unfolded, and a total weight of approximately 9.5 kg. With a total takeoff weight of 15 kg, the UAV was able to carry a payload of approximately 5.5 kg. Operation of the UAV was handled via a remote controller with a 2.4 GHz radio link and maximum transmission distance of approximately 5 km coupled with a tablet computer, allowing for autonomous flight using DJI Ground Station Pro software. In addition to the GPS system incorporated into the imaging system, location information is supplied through the onboard flight control system on the UAV via three additional GPS antennae.

### 2.4. Flight Overview

All UAV flights were conducted either by, or under the supervision of, a pilot certified through the Federal Aviation Administration (FAA) Part 107 licensure during each data collection campaign on 17 August and 9 September 2021. The airspace above each sampling site was verified as Class D using the FAA-certified AirMap software (AirMap, Santa Monica, CA, USA). To minimize changes in lighting conditions, flights were scheduled within two hours of solar noon, or between approximately 11:00–15:00 MDT on mostly clear days with calm wind to minimize image blurring and errors in calibration techniques caused by tilting of the UAV. Prior to flying, both the UAV and airborne imaging system were inspected and calibrated according to manufacturer specifications [58,59] to ensure proper operation and precise heading and location information. After inspection and calibration, reflectance tarps were laid out along the river bank within the field of view of the imaging system.

Flight plans were created prior to each mission using DJI Ground Station Pro software. Each flight was conducted in 300–500 m straight-line segments over the center of the river channel at an above ground level of 120 m, the maximum allowed by the FAA, and a flight speed of approximately 2.5 m/s. Flight parameters were chosen to maximize the amount of river channel captured in each image while minimizing image distortion caused by increased flying speeds and frequent aerial movements.

The airborne hyperspectral imaging system and downwelling irradiance sensor were configured prior to each flight using Resonon Ground Station software. Exposure settings on the imager were held constant throughout each flight after being set through an auto exposure routine using the 11% reflectance target illuminated by direct sunlight. Frame rate was calculated by inputting the flight height and speed into the Ground Station software. The pushbroom hyperspectral imager was set to capture a new data cube after 2000 image lines were captured, at which time the spectrometer would capture a downwelling spectral irradiance measurement.

### 2.5. Ground-Truth Data Collection

To build a relationship between image data collected by the hyperspectral imager and algal pigment abundance, ground-truth data were collected immediately prior to UAV flights on 17 August and 9 September 2021. On 17 August, prior to UAV flights, seven plots were established randomly on the stream bottom at each sample site. Circular plots were delineated with 112 cm sections of white hose filled with sand to ensure that spectral and pigment data were derived from the same locations. Sampling on 9 September was conducted at the Gold Creek site with 20 plots placed in four sets of five organized transects, purposefully moving from areas of low to high growth (Figure 2).

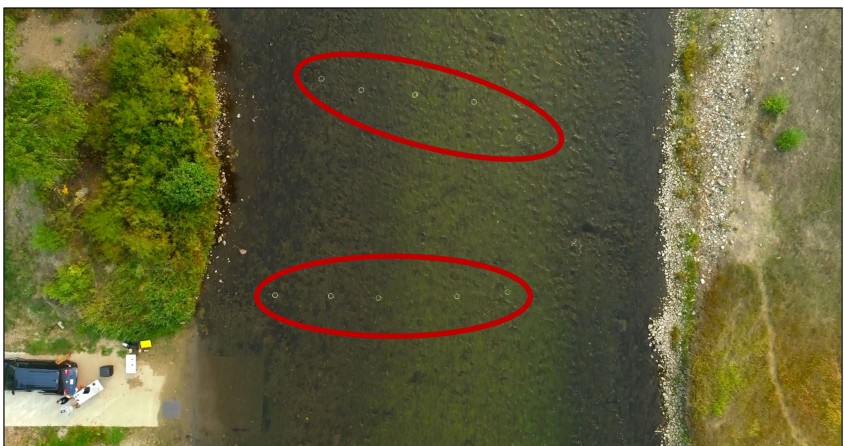

**Figure 2.** Sample plots and transects at the Gold Creek site on 9 September 2021. Five hoops (small white circles) were placed along four transects (red ovals, two of five shown) to establish plots moving from areas of low to high algal growth.

After image data were collected, water depth was measured and all benthic substrata (rocks, etc.) were collected within each plot. Pigment abundance was determined as areal standing crops for three growth forms: filamentous algae, epiphytes, and epilithon. Filamentous algae were removed from the surface of stones within each plot, rinsed and squeezed, and retained independently. Rinse from cleaning (i.e., squeezing and rinsing) filamentous algae was retained separately to quantify epiphytic pigments. Epilithic algae were removed using steel brushes and the resultant slurry was processed following Steinmann et al. [13]. All samples were bagged and placed in a cooler to avoid direct sunlight and keep samples chilled.

Samples were analyzed for chl *a* using 90% buffered acetone for pigment extraction and spectrophotometric assessment with acidification following the Ritchie method [60]. Phycocyanin standing crops extracted from epilithic and epiphytic samples were obtained through the use of a handheld fluorimeter (Aquafluor 8000, Turner Designs, Sunnyvale, CA, USA) following Moulton et al. [61] and Cremella et al. [62]. Calculations for final standing crops included corrections for sub-sampling and were normalized to sampling area.

For each plot, benthic organic matter (BOM) standing stocks were measured for each growth form as AFDM ($g/m^2$) following standard protocols [13]; dry mass was measured following 24 h at 60 °C, and ash mass after sample ignition for one hour at 550 °C, cooling and re-weighing. For slurries of epilithon, subsamples were filtered through pre-weighed Whatman GF/F (0.7 μm pore size) filters. For filamentous algae, subsamples of wet weight were used for drying and combustion. One corrupted data point was removed from the 17 August Gold Creek data set.

*2.6. Image Preprocessing*

2.6.1. Image Calibration

Hyperspectral data captured by the imaging system have a native pixel format of raw digital counts. Calibrating the pixel values to a common physical scale is key to achieving accurate image processing and comparison among imaging systems. Image calibration was performed using two methods: reflectance targets laid out along the river bank and a downwelling irradiance sensor mounted on the UAV.

The first of these methods used two G8T portable fabric reflectance targets (Group 8 Technology Inc., Provo, UT, USA). The reflectance standards had spectral reflectances of approximately 6% and 11% from 400–1000 nm, with sizes of 1.2 × 1.2 m and 2 × 2 m, respectively. Both tarps were laid out in flat areas along the river bank in locations that would be captured by the imager during UAV flights. As the tarps were contained in only a

portion of a single image, the pixels containing the tarps were used as a reflectance standard from which all other image pixels were calibrated throughout the flight using Equation (1):

$$\rho(\lambda)_{scene} = \frac{L(\lambda)_{scene} - L(\lambda)_{dark}}{L(\lambda)_{tarp} - L(\lambda)_{dark}} \rho(\lambda)_{tarp}, \tag{1}$$

where $\rho(\lambda)_{scene}$ is the spectral reflectance of the scene, $L(\lambda)_{scene}$ is the spectral radiance of the imaged scene, $L(\lambda)_{tarp}$ is the spectral radiance of the reflectance standard, $L(\lambda)_{dark}$ is the dark current signal on the image sensor, and $\rho(\lambda)_{tarp}$ is the spectral reflectance of the standard. Though this method is an accepted calibration technique, it relies on several assumptions, such as constant lighting conditions throughout the flight and pixel-to-pixel response consistency.

The second method removed the assumptions implicit in the first method by using an Ocean Insight Flame UV-VIS spectrometer (Ocean Insight, Orlando, FL, USA) fitted with a cosine corrector and calibrated to measure downwelling spectral irradiance from 400–1000 nm with nominal spectral resolution of 1.34 nm. The spectrometer was configured to synchronously measure spectral downwelling irradiance with each hyperspectral data cube captured by the imager. This method allowed each hyperspectral data cube to be converted to reflectance with precise illumination information throughout the flight and thereby removed the assumption of unchanging illumination, using

$$\rho(\lambda)_{scene} = \frac{L(\lambda)_{scene} - L(\lambda)_{dark}}{\frac{E(\lambda)_{downwelling}}{\pi} - L(\lambda)_{dark}}, \tag{2}$$

where $E(\lambda)_{downwelling}$ is the downwelling spectral irradiance and $\pi$ is the hemispheric projected solid angle.

### 2.6.2. Pixel Selection

After the hyperspectral images were converted from digital counts to reflectance, spatial pixels within each plot that were not saturated by sun glint were averaged using Resonon Spectronon software. For a 120 m flight altitude with a 17 mm focal length lens, the across-track image contained 900 pixels in a 17.6-degree field of view, such that each plot contained an average of approximately 40 pixels after flying over each plot. The spectra contained in each of these spatial pixels was averaged to create a single spectrum for each plot.

### *2.7. Image Processing*

Hyperspectral image data were processed by isolating the signal reflected from the stream bed harboring algal growth, converting image data to physical units of reflectance, and developing and applying spectral indices following the steps outlined below.

### 2.7.1. Theoretical Basis for Isolating Stream Bed Reflectance

The goal of the image processing method presented was to relate the upwelling radiance measured by the hyperspectral imaging system to the pigment abundance contained within benthic algal blooms. Following a development similar to that presented by Legleiter, et al. [63,64], total upwelling spectral radiance [W/(m$^2$ sr nm)] measured by the UAV-based imaging system can be written as

$$L_{tot}(\lambda) = L_b(\lambda) + L_{wb}(\lambda) + L_{ws}(\lambda) + L_{atm}(\lambda), \tag{3}$$

where $L_{tot}$ is the total upwelling spectral radiance, $L_b$ is the spectral radiance reflected from the benthos, $L_{wb}$ is the spectral radiance scattered to the imager from the water body, $L_{ws}$ is the spectral radiance reflected from the water's surface, $L_{atm}$ is the spectral radiance scattered from the atmosphere, and all terms are a function of wavelength, $\lambda$. Each term can also have variable polarization, which creates radiometric errors if the polarization-sensitive instrument response is ignored [65,66]; however, this becomes negligible for the small incidence angles (less than 8°) employed in this work. Of the terms addressed in

Equation (3), $L_b$ is of primary interest when relating image data to a difference in benthic algal pigment contents. Owing to the nature of the UCFR and UAV-based data collection, several assumptions can be made to simplify Equation (3) and isolate the signal of interest.

With the exception of stormflow and snowmelt conditions, the UCFR is clear and shallow, meaning the majority of the downwelling solar irradiance will be reflected from the stream bed itself, making $L_b$ the dominant signal reaching the imager. Upwelling radiance from optical scattering within the water body prior to reaching the bottom substrate, $L_{wb}$, largely depends on the water's inherent optical properties. These generally include the absorption coefficient, scattering coefficient, and volume scattering function, which are functions of both depth and dissolved and suspended matter present in the stream [67,68]. Owing to the clarity and shallowness of the UCFR, radiance generated through $L_{wb}$ can be considered negligible when compared to the relatively large signal of $L_b$. The signal generated by $L_{ws}$ largely depends on the reflectance of the water's surface, which is a function of surface conditions and stream morphology. The reflectance of the water's surface has been measured as low as 0.03 between 400–800 nm, with reflectance values approaching 0 for calm surfaces [63], making $L_{ws}$ small compared to $L_b$. The final term, $L_{atm}$, typically becomes negligible when optical signals take relatively short paths through a clear atmosphere, such as UAV operation at 120 m on a clear day. However, under smoky conditions, which often occur during the summer in Montana, atmospheric radiance may increase. Radiative transfer simulations show that elevated atmospheric scattering driven by smoke generates a spectrally-flat reduction in contrast from 400–1000 nm; however, the increased $L_{atm}$ is still much smaller than the signal generated by $L_b$.

Taking these assumptions and simplifications into account, total upwelling radiance reaching the imager can be considered to be dominated by upwelling radiance from the benthos and Equation (3) can be rewritten only in terms of $L_b$ as

$$L_{tot} \approx L_b = \frac{E_d T_{aw} T_{wa} R_b e^{-2\kappa d}}{\pi}, \tag{4}$$

where $E_d$ is the downwelling irradiance [W/m$^2$]; $T_{aw}$ is the air-water interface transmittance; $T_{wa}$ is the water-air interface transmittance; $R_b$ is the reflectance of the stream bed; $\kappa$ is the optical extinction coefficient; $d$ is the depth of the water body, which must be considered on both its downwelling and upwelling path through the water; and the factor of pi represents the bidirectional reflectance distribution function for a Lambertian surface. Transmission across the air-water interface occurs twice, once when the optical signal enters the water ($T_{aw}$) and again when the scattered sunlight leaves the water ($T_{wa}$). For small incidence angles, the transmittance across this interface is equal when moving between the media; however, for large incidence angles these terms may need to be considered separately.

As the primary signal of interest for identifying algal pigment standing crops is the stream bed reflectance, Equation (4) can be rewritten as

$$R_b = \frac{R_{tot}}{T_{aw} T_{wa} e^{-2\kappa d}}, \tag{5}$$

where $R_{tot}$ is the reflectance measured at the imager, or $L_{tot}/E_d$. Equation (5) provides a basis for isolating the reflectance of the stream bed to accurately assess changes in pigment abundance.

Calculating or measuring the extinction coefficient, $\kappa$, enables correction for the effects of optical absorption and scattering within the water as a function of wavelength and depth. Accounting for these losses may allow for the use of wavelengths that are strongly absorbed by water, which contain important pigment information. However, correcting for the effects of water extinction requires continuous measurement of the depth of the studied water body, making the method largely inaccessible for most remote sensing applications. The results presented here do not include an extinction correction because it is less important for shallow water and it is preferred to present a method that is more readily implemented by others using the equipment presented herein.

2.7.2. Development of Spectral Indices and Application to Algal Standing Crops

Using methods similar to those presented in earlier works [36,64], a brute-force analysis was performed to relate hyperspectral reflectance data to in situ samples of chl *a* and phycocyanin. Reflectance data across the spectral range of 400–850 nm from each measurement plot were extracted to form all possible band ratio combinations of the form $R_{\lambda_1}/R_{\lambda_2}$, where $R_\lambda$ represents the reflectance at a single 2.1 nm-wide spectral channel. Data were omitted for wavelengths longer than 850 nm, where the signal-to-noise ratio declines rapidly because of increasing water absorption and decreasing detector responsivity. The resulting values were matched to corresponding in situ pigment measurements and linear regressions performed to relate pigment standing crops to each band ratio's spectral signal. The goodness of fit for each model was assessed via the coefficient of determination ($R^2$) for each data set. The approach generated a large number (45,796) of band ratios and $R^2$ values for each data set analyzed. Data sets containing the band ratios and the resultant $R^2$ values were analyzed with frequency distributions to characterize the spread of potentially useful band ratios. A matrix of $R^2$ values for each band ratio combination was developed to depict correlation intensity across all band ratio combinations. The matrix was used to highlight areas sensitive to the target pigment, from which the band ratio with the highest $R^2$ value was saved as the optimal band ratio for the data set.

With this approach, two major factors influencing the identity of the appropriate spectral signal (i.e., band ratio) and the quality of the linear models employed to estimate pigment standing crops were addressed. First, independent models were generated for different growth forms including: (i) filamentous plus epiphytic (fila/epip), (ii) epilithic, and (iii) total (filamentous, epiphytic, and epilithic). Models estimating phycocyanin abundance were restricted to epiphytic and epilithic growth, as no bluegreen bacteria were associated with *Cladophora* filaments. Second, the influence of RAB development over time was addressed by comparing combined and independent models derived from multiple dates (17 August, 9 September). Finally, the combined influence of form and stage was analyzed by generating models specific to individual growth forms for different sampling dates.

*2.8. Uncertainty Analysis*

The optimal band ratios selected by the brute-force method were based on correlating a small number of pigment observations (n = 33) to spectral models with a large number of potential explanatory variables (n = 45,796), creating conditions that could generate spurious relationships. To test the generalizability and uncertainty around all band ratios, including those identified as optimal by the brute-force method, a resampling approach was used. In this approach, the 33 pigment observations were subsampled into groups of 26 (representing 80% of the observations) and these subsamples were repeated 1000 times without replacement. This resampling approach was chosen as it would allow comparison between consistently sized subsamples of the data. For each subsample, a linear regression was performed between each of the 45,796 possible band ratios and the subsample of pigments, generating 45,796,000 linear models and $R^2$ values (1000 for each band ratio). The average $R^2$ value was calculated for each band ratio across all 1000 resamples. With this method, the performance of each of the optimal band ratios identified by the brute-force analysis could be assessed against each pigment subsample, giving a distribution of $R^2$ values and a standard deviation around the mean $R^2$ value for each pigment growth form. The subsampling analysis was run for total standing crops of chl *a* and phycocyanin, as well as after separating by growth form.

**3. Results**

*3.1. RAB Characteristics*

When plots were randomly established at both Gold Creek and Bear Gulch, total chl *a* standing crop ranged from a high of 316.5 mg/m$^2$ to a low of 69.3 mg/m$^2$ and averaged 191.4 ± 33.1 mg/m$^2$ and 183.5 ± 31.8 mg/m$^2$ at Gold Creek and Bear Gulch, respectively

(Table 1). When samples at Gold Creek were purposefully distributed to include transects into the primary RAB, total chl *a* standing crops ranged from a high of 377.8 mg/m$^2$ to a low of 46.1 mg/m$^2$ and averaged 186.0 $\pm$ 30.3 mg/m$^2$. The majority of chl *a* was found in standing crops of filamentous growth across all field sites and sampling dates. Epilithic chl *a* standing crops tended to be greater than those of epiphytic chl *a* at Gold Creek, whereas Bear Gulch contained similar levels of each. Total standing crop of BOM was highest on 17 August at Gold Creek (97.4 g AFDM/m$^2$), including a large proportion (88%) of filamentous growth. Between 17 August and 9 September, total BOM decreased at Gold Creek, largely due to a decrease in filamentous forms. Bear Gulch showed the lowest standing stocks of filamentous BOM, but larger standing stocks of epiphytic and epilithic BOM compared to the Gold Creek data sets. Total phycocyanin standing crops were similar between field sites; however, abundance was distributed almost evenly between epiphytic and epilithic growth forms at Bear Gulch whereas the majority of phycocyanin standing crops were contained in epilithon at Gold Creek.

**Table 1.** Standing crop of chlorophyll a (chl *a*), benthic organic matter (BOM), and phycocyanin at Gold Creek and Bear Gulch. Data are means $\pm$ standard error; n = 6 at Gold Creek on 17 August, n = 7 at Bear Gulch on 17 August, and n = 20 at Gold Creek on 9 September.

| Field Site | Field Date | Growth Form | chl *a* (mg/m$^2$) | BOM (g FDM/m$^2$) | Phycocyanin (µg/m$^2$) |
|---|---|---|---|---|---|
| Gold Creek | 17 August | Filamentous | 158.9 $\pm$ 30.3 | 85.8 $\pm$ 13.3 | NA |
| | | Epiphytic | 7.3 $\pm$ 2.8 | 4.2 $\pm$ 0.4 | 6.5 $\pm$ 2.2 |
| | | Epilithic | 25.2 $\pm$ 2.2 | 8.8 $\pm$ 13.9 | 31.3 $\pm$ 4.6 |
| | | Total | 191.4 $\pm$ 33.1 | 97.4 $\pm$ 13.9 | 37.8 $\pm$ 4.6 |
| | 9 September | Filamentous | 149.5 $\pm$ 24.0 | 67.4 $\pm$ 11.0 | NA |
| | | Epiphytic | 10.0 $\pm$ 2.5 | 5.0 $\pm$ 1.7 | 7.2 $\pm$ 2.4 |
| | | Epilithic | 26.6 $\pm$ 3.8 | 9.9 $\pm$ 1.3 | 27.2 $\pm$ 4.6 |
| | | Total | 186.0 $\pm$ 30.3 | 82.4 $\pm$ 12.1 | 34.4 $\pm$ 5.5 |
| Bear Gulch | 17 August | Filamentous | 124.8 $\pm$ 25.8 | 55.5 $\pm$ 14.8 | NA |
| | | Epiphytic | 26.8 $\pm$ 9.9 | 17.2 $\pm$ 3.7 | 20.8 $\pm$ 5.2 |
| | | Epilithic | 31.9 $\pm$ 5.2 | 15.4 $\pm$ 2.5 | 21.7 $\pm$ 2.6 |
| | | Total | 183.5 $\pm$ 31.8 | 88.1 $\pm$ 18.1 | 42.5 $\pm$ 5.3 |

### 3.2. Spectral Models

Estimative modeling began by running the brute-force analysis using the combined data set, composed of pigment data collected across all field sites and dates and averaged spectra from corresponding plots. Each plot consisted of a single spectrum obtained by averaging the spectra contained in all hyperspectral pixels within the plot (Figure 3). The model was run using total chl *a* standing crops (i.e., sum of all growth forms) and total phycocyanin abundance, generating three output products (Figure 4): a heat map of coefficient of determination ($R^2$) values for every possible band ratio (Figure 4a), a histogram showing the counts of $R^2$ values (Figure 4b), and a linear regression between the optimal band ratio and the targeted pigment (Figure 4c).

The heat maps for each regression represent the resulting $R^2$ value generated for every possible band ratio, created from a particular wavelength as denominator (y-axis) and numerator (x-axis). The heat maps allow for a quantifiable means of selecting the optimal band ratio (i.e., The band ratio with the highest $R^2$ value) and determining spectral regions which are sensitive to the targeted pigment. Next, the histograms show the salience of the regressions generated by the brute-force analysis by displaying the number of band ratios which generated the $R^2$ values shown in the heat maps. Finally, the linear regressions for each data set are a graphical representation of the fit between the best-performing band ratio, selected by the brute-force algorithm, and the pigment under analysis. For each linear

regression, the field location and date from which the ground-truth pigment standing crop was measured is designated with a unique marker and color.

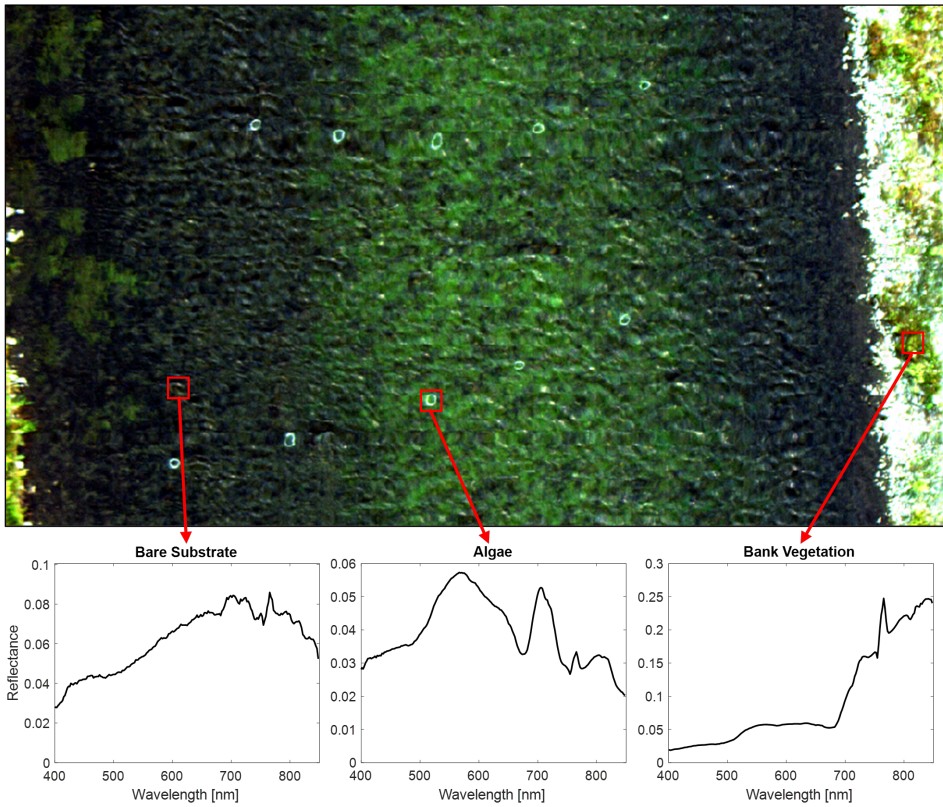

**Figure 3.** RGB image created from one of the hyperspectral image cubes gathered using the UAV system at Gold Creek. Examples of the complete spectra contained at each pixel are shown for bare river substrate, *Cladophora* algae, and bank vegetation for the spectral range of 400–850 nm. Note the varying reflectance axes and strong water absorption beyond approximately 725 nm that suppresses the strong near-infrared reflectance of the submerged algae.

A band ratio of 684/674 nm was selected as the optimal ratio for estimating chl *a* standing crops, which is likely based on the known chl *a* absorption line near 665 nm [69,70]. Of the 45,796 possible band ratios, the optimal band ratio was one of eight which produced an $R^2$ above 0.40 (Figure 4b). Analysis of the combined data set showed promising results when estimating chl *a* standing crops, with an $R^2$ = 0.57 and root-mean-square error (RMSE) = 66.29 mg/m$^2$. Regression analysis of phycocyanin standing crops resulted in worse estimation performance, with an $R^2$ = 0.41 and RMSE = 15.54 mg/m$^2$. An optimal band ratio of 753/824 nm was selected, highest among 10 band ratios which resulted in an $R^2$ above 0.35 (Supplementary Figure S1).

### 3.2.1. Separation by Growth Form

Though the combined data set showed promising results, the optical signal received by the imaging system was likely dominated by either large mats of filamentous algae obscuring the epilithic growth or areas of solely epilithic growth with minimal filamentous algae. To explore this possibility, the pigment abundances related to the filamentous algae and epiphytes were separated from the abundances generated from epilithic sources, and each data set was used as input to the brute-force analysis using the same imagery, with zero-valued points removed. As epiphytes tend to collect and grow on the surface of filamentous algae, the optical signals for these growth forms were likely conflated; therefore, the chl *a* standing crops associated with these two forms were combined into a single group.

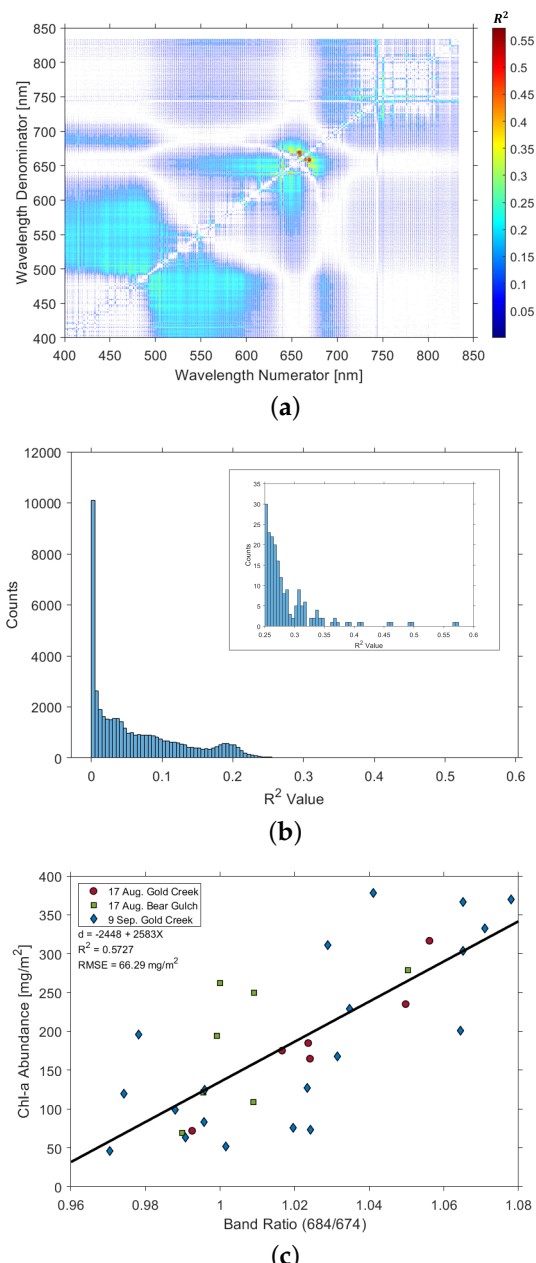

**Figure 4.** Analytics from the regression analysis of total chl *a* standing crops (sum of filamentous, epiphytic, and epilithic sources) from the combined (all field sites) data set; (**a**) Heat map of $R^2$ values generated by fitting each band combination against total chl *a* standing crops. Colors within the heat map represent the $R^2$ value, with wavelength numerators and denominators shown on the x- and y-axes, respectively; (**b**) Histogram of the $R^2$ values generated using the brute-force analysis. The insert shows counts beyond an $R^2$ = 0.25, showing that relatively few band ratios generate fits in this region; (**c**) Linear regression analysis between chl *a* abundance (y-axis) and optimal band ratio (x-axis). Data collected on 17 August 2021 at the Gold Creek and Bear Gulch site represented by red circles and green squares, respectively, while data collected on 9 September 2021 are shown with blue diamonds.

Filamentous Plus Epiphytic

After separating each data set, the strength of relationships between target pigment abundance and image data were compared to the combined data set's performance when estimating total pigment standing crops. The difference in model performance is described as $\Delta R^2$, which is positive for increases in $R^2$ and negative for decreases. The performance

of the regression analysis modestly improved when estimating fila/epip chl *a* standing crops individually, with a $\Delta R^2$ of 0.05, increasing from 0.57 to 0.62, while maintaining an optimal band ratio of 684/674 nm. With improved performance, the salience of this ratio also increased; the optimal band ratio was one of six generating an $R^2$ above 0.50 (Supplementary Figure S2).

Estimation of phycocyanin standing crops was limited to epiphytic and epilithic growth forms as phycocyanin was not associated with the filamentous growth forms present in the UCFR. Estimative performance of epiphytic phycocyanin standing crops decreased compared to the combined data set estimates, with a $\Delta R^2$ of $-0.08$. The decreased performance was associated with an optimal band ratio similar to those selected for chl *a* estimation, with a band ratio of 688/655 nm (Supplementary Figure S3).

Epilithic

Estimation of epilithic chl *a* standing crops decreased compared to the combined data set, with a loss in $R^2$ resulting in a $\Delta R^2$ of $-0.13$, decreasing from 0.57 to 0.44 while the optimal band ratio shifted to longer wavelengths of 729/809 nm (Supplementary Figure S4). Performance for estimating epilithic phycocyanin standing crops improved compared to the combined data set, with $R^2$ values increasing from 0.41 to 0.50, resulting in a $\Delta R^2$ of 0.09 while using an optimal band ratio of 738/804 nm (Supplementary Figure S5).

The improved strength of the relationship for estimating chl *a* in fila/epip growth forms separately and decrease in performance for epilithic chl *a* suggests that the optical signal from filamentous and epiphytic algal growth dominates the imaged scene during a large bloom. The increased performance for epilithic phycocyanin is likely due to the larger abundance of phycocyanin commonly found in epilithic algal growth. Those pixels which contain epilithic growth, which are not obscured by filamentous growth, are likely better estimated when separated by growth form.

3.2.2. Separation by Phenology

Sources of algal biomass change throughout the summer growing season, with the ratio of biomass sourced from epilithon, epiphytes, and filamentous algae shifting as a function of stream conditions. This change has been observed to depend on sampling location along the UCFR, with field sites farther downstream progressing through their life cycle prior to those closer to the headwaters. Compared to both Gold Creek sites, the Bear Gulch site visually appeared to contain elevated epiphyte colonization relative to filamentous organic matter. To explore the role phenology had in band ratio selection, the data collected from the two field sites were separated, forming a Bear Gulch data set (n = 7) and Gold Creek data set (n = 26). These two data sets were used as inputs to the brute-force analysis to estimate standing crops from all pigment sources, epiphytic and filamentous sources, and epilithon.

Bear Gulch
Total

Separation based on phenology led to improvements in the overall strength of the regression relationship, with performance increasing for all pigments and sources. When estimating total chl *a*, the brute-force analysis selected a band ratio of 554/536 nm, resulting in a $\Delta R^2$ of 0.38, increasing from the combined model's $R^2$ value of 0.57 to 0.95 (Supplementary Figure S6). Estimation of total phycocyanin at Bear Gulch shifted the optimal band ratio from 753/824 nm in the combined model to 625/619 nm (Supplementary Figure S7). The associated $\Delta R^2$ was 0.51, with $R^2$ values increasing from 0.41 to 0.92.

Filamentous plus Epiphytic

Improvements in the estimation of the fila/epip and epilithic groups were also seen after separating the data by location. When estimating fila/epip chl *a* at Bear Gulch, the

brute-force analysis selected a shorter wavelength ratio of 554/536 nm with a $\Delta R^2$ of 0.39 associated with an $R^2$ of 0.96 (Supplementary Figure S8). Similarly, the estimation of epiphytic phycocyanin improved, with a $\Delta R^2$ of 0.53, after $R^2$ values increased from 0.41 to 0.94 and selecting an optimal band ratio of 748/740 nm (Supplementary Figure S9).

Epilithic

Epilithic pigment estimation accuracy also increased for both chl *a* and phycocyanin, with $R^2$ values improving from 0.57 to 0.86 and 0.41 to 0.89, respectively, leading to $\Delta R^2$ values of 0.29 and 0.48 compared to the combined model (Supplementary Figure S10). Estimation of epilithic chl *a* from the Bear Gulch data set relied on shorter-wavelength ratios as compared to the combined data set, selecting a band ratio of 573/589 nm. The ratios selected for epilithic phycocyanin also shifted to shorter wavelengths, using a band ratio of 525/519 nm (Supplementary Figure S11).

Gold Creek
Total

Analyzing the Gold Creek data separately resulted in improved estimates compared to the combined data set across all growth forms. The strength of the relationship between total chl *a* and image data at Gold Creek improved from an $R^2$ of 0.57 to 0.63 ($\Delta R^2$ = 0.06) while for total phycocyanin, $R^2$ values increased from 0.41 to 0.46, resulting in a $\Delta R^2$ of 0.05. The optimal band ratio for estimating total chl *a* from the Gold Creek data set remained the same between the data sets, maintaining a band ratio of 684/674 nm (Supplementary Figure S12). The optimal band ratio for total phycocyanin showed a near-inverse relationship compared to the combined data set, shifting from 753/824 nm to 804/748 nm (Supplementary Figure S13).

Filamentous plus Epiphytic

The strength of the linear regression was highest at Gold Creek when estimating fila/epip chl *a*, with a $\Delta R^2$ of 0.11, after $R^2$ values increased from 0.57 to 0.68 (Figure 5). Notably, the optimal band ratio remained the same for the combined data set and the Gold Creek data set. This band ratio is the vicinity of known chl *a* absorption near 665 nm, suggesting the brute-force analysis relied on a change in chl *a* abundance when estimating total and fila/epip chl *a* standing crops. Estimation of epiphytic phycocyanin also showed modest improvements, increasing from an $R^2$ of 0.41 to 0.43, an improvement associated with a $\Delta R^2$ of 0.02 and the selection of a band ratio of 778/774 nm (Supplementary Figure S14).

Epilithic

The regression analysis for estimating epilithic pigmentation also improved for chl *a* (Supplementary Figure S15) and phycocyanin (Supplementary Figure S5), with $R^2$ values increasing from 0.57 to 0.62 and 0.41 to 0.59, respectively, leading to $\Delta R^2$ values of 0.05 and 0.18. The optimal band ratio for estimating epilithic phycocyanin shifted to longer wavelengths, selecting an optimal band ratio of 804/740 nm.

The correlation maps for the Gold Creek data set show that wavelengths surrounding chl *a* absorption lines are favored when estimating total chl *a* abundance, fila/epip chl *a*, and epiphytic phycocyanin. However, longer-wavelength band ratios are preferred when estimating the remaining pigments and forms, often selecting wavelengths over 700 nm.

### 3.2.3. Results Summary

Separating the combined data set by growth form (pigment source) and phenological state (field site) generally resulted in improved performance for estimating both chl *a* (Table 2) and phycoycanin (Table 3) standing crops, with the exception of epilithic chl *a* for which performance dropped after growth form separation. The strongest relationship between band ratio and pigment abundance was found after separating by phenological state and estimating pigment standing crops at the Bear Gulch. Although the linear

regression models at the Bear Gulch site produced $R^2$ values above 0.86, this field site was limited to seven data points, which may introduce problems such as spurious fits. Optimal band ratio selection was fairly robust when estimating chl *a* abundance at Bear Gulch, converging on a band ratio of 554/536 nm for total and fila/epip chl *a*; however, optimal band ratio selection for phycocyanin abundance was more variable.

**Table 2.** Model performance and optimal band ratio selected for chl *a* abundance after separating data sets by growth form and phenological state. The $\Delta R^2$ value describes the change in $R^2$ after separating the data set, relative to the performance of the combined data set estimating total pigment abundance.

| Data Set | Growth Form | Optimal Band Ratio | $R^2$ | $\Delta R^2$ |
|---|---|---|---|---|
| Combined | Total | 684/674 | 0.57 | - |
|  | Fila/epip | 684/674 | 0.62 | 0.05 |
|  | Epilithic | 729/809 | 0.44 | −0.13 |
| Bear Gulch | Total | 554/536 | 0.95 | 0.38 |
|  | Fila/epip | 554/536 | 0.96 | 0.39 |
|  | Epilithic | 573/589 | 0.86 | 0.29 |
| Gold Creek | Total | 684/674 | 0.63 | 0.06 |
|  | Fila/epip | 684/674 | 0.68 | 0.11 |
|  | Epilithic | 804/738 | 0.62 | 0.05 |

**Table 3.** Model performance and optimal band ratio selected for phycocyanin abundance after separating data sets by growth form and phenological state. The $\Delta R^2$ value describes the change in $R^2$ after separating the data set, relative to the performance of the combined data set estimating total pigment abundance.

| Data Set | Growth Form | Optimal Band Ratio | $R^2$ | $\Delta R^2$ |
|---|---|---|---|---|
| Combined | Total | 753/824 | 0.41 | - |
|  | Epiphytic | 688/655 | 0.33 | −0.08 |
|  | Epilithic | 738/804 | 0.50 | 0.09 |
| Bear Gulch | Total | 625/619 | 0.92 | 0.51 |
|  | Epiphytic | 748/740 | 0.94 | 0.53 |
|  | Epilithic | 525/519 | 0.89 | 0.48 |
| Gold Creek | Total | 804/748 | 0.46 | 0.05 |
|  | Epiphytic | 778/774 | 0.43 | 0.02 |
|  | Epilithic | 804/740 | 0.59 | 0.18 |

The strongest relationship using the larger Gold Creek data set was found after separating the data sets by growth form and analyzing epilithic phycocyanin and fila/epip chl *a* standing crops (Figure 5). Estimation of epilithic phycocyanin abundance produced an $R^2$ of 0.59 and utilized a band ratio of 804/740 nm (Figure 5b,d,f). Analysis of fila/epip chl *a* produced an $R^2$ of 0.68 and converged on the optimal band ratio of 684/674 nm, found in several chl *a* abundance measurements for the Gold Creek and combined data sets (Figure 5a,c,e). When applied at Bear Gulch, the optimal band ratio for the combined and Gold Creek data produced reasonable fits when estimating chl *a* standing crops. Fitting to total and fila/epip chl *a* standing crops, the band ratio of 684/674 nm produced $R^2$ values of 0.37 and 0.36, respectively, meaning this band ratio may carry meaningful information across field sites (i.e., phenological states).

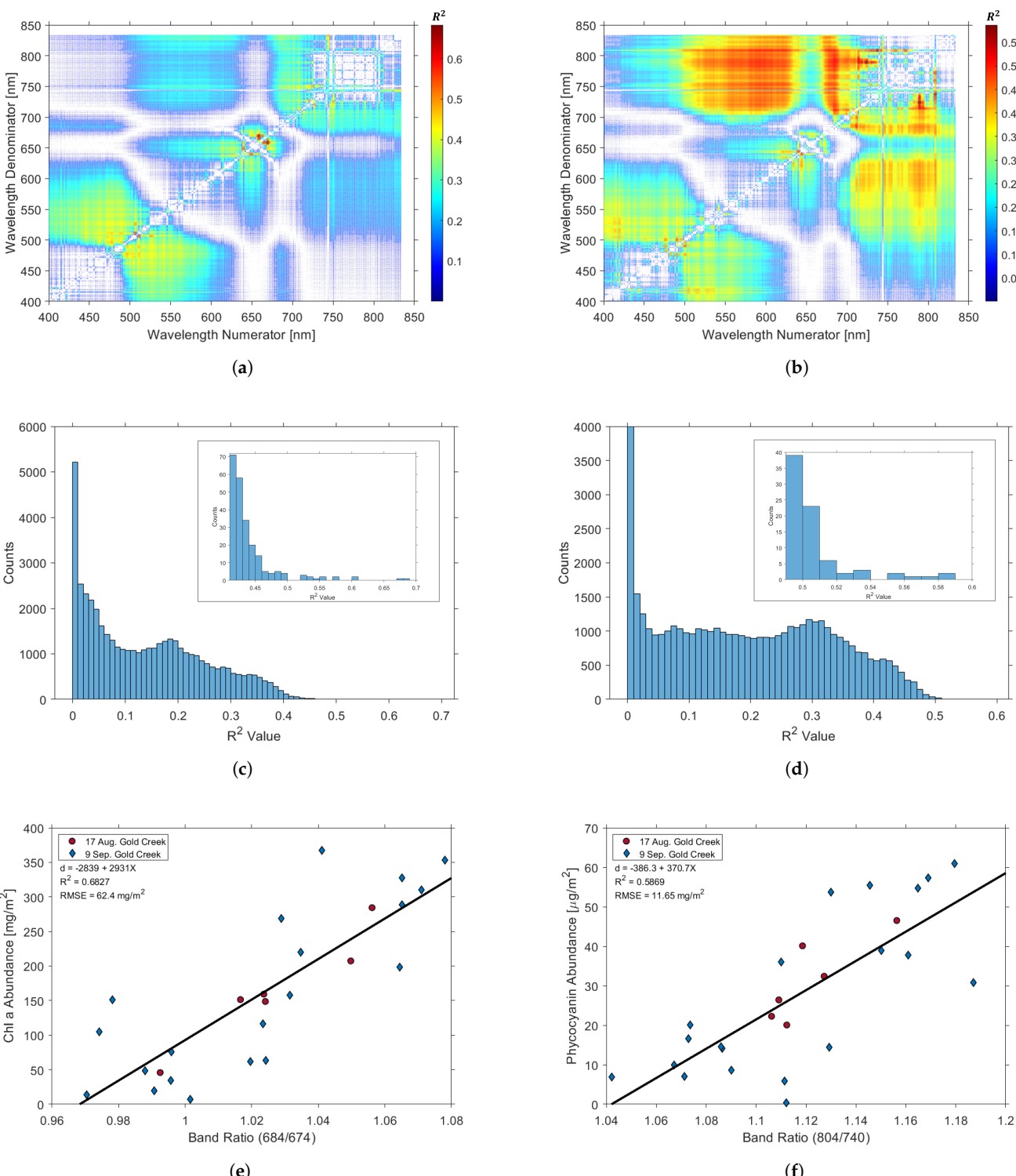

**Figure 5.** Linear regressions, correlation maps, and histograms for the Gold Creek data set. Results gathered via the brute-force analysis for estimating fila/epip chl *a* abundance (left column) and epilithic phycocyanin abundance (right column); (**a**) Gold Creek fila/epip chl *a* correlation map; (**b**) Gold Creek epilithic phycocyanin correlation map; (**c**) Gold Creek fila/epip chl *a* histogram; (**d**) Gold Creek epilithic phycocyanin histogram; (**e**) Gold Creek fila/epip chl *a* linear regression; (**f**) Gold Creek epilithic phycocyanin linear regression.

Due to its strong performance and frequent selection as optimal, the linear equation obtained from the band ratio of 684/674 was used to estimate pigment abundances for the top-performing data set, found for fila/epip chl *a* measurements at the Gold Creek field site, and compared to the measured abundances (Figure 6). In general, the model showed promising results, but tended to overestimate low chl *a* abundances (below $\approx 100$ mg/m$^2$) and underestimate elevated abundances (above $\approx 250$ mg/m$^2$).

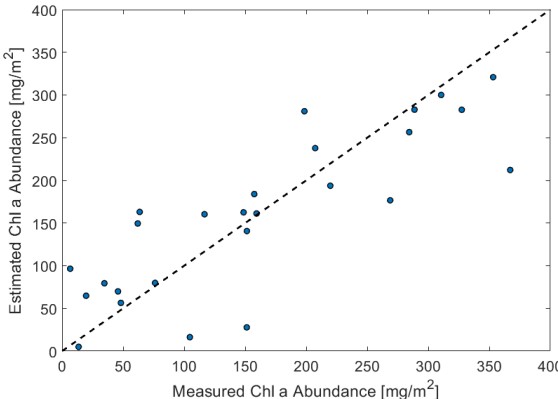

**Figure 6.** Estimated versus measured fila/epip chl *a* abundance at the Gold Creek field site using the regression analysis generated by the brute-force method. Points falling along the 1:1 line (dashed) represent perfect performance.

### 3.3. Generalizability of Band Ratios

The resampling approach identified the same optimal band ratios and similar average $R^2$ values as the brute-force method for four out of the six combinations of pigment and growth form, and identified the inverse band ratio for one. For all of the chl *a* growth forms, the same optimal band ratios were identified by both the brute-force method and resampling 80% of the observations 1000 times, and the mean $R^2$ value from the resampling approach was identical to that identified by the brute-force method on all of the samples (Table 4). Additionally, the resulting data distribution of the $R^2$ values across all 1000 resamples for the optimal band ratios were tightly constrained (Figure 7), with standard deviations between 0.05 and 0.06. For phycocyanin, the resampling approach selected the same optimal band ratio for total abundance, the inverse band ratio for epilithic (i.e., brute force, 738/804; resampling, 804/738), while both band ratios had $R^2$ values that were nearly identical between the resampling and brute-force approaches and small standard deviations. In contrast, the resampling approach identified a different optimal band ratio for the epiphytic growth form for phycocyanin (i.e., brute force, 688/655; resampling, 684/674) and the $R^2$ value was higher for the resampling approach for this alternative band ratio. If the $R^2$ is examined from the resampling approach for the optimal band ratio identified by the brute-force method, a mean $R^2$ of $0.18 \pm 0.19$ is generated.

**Table 4.** Optimal band ratio, mean $R^2$ value, and standard deviation identified after running the uncertainty analysis on total pigment abundance and after separating by growth form.

| Pigment | Growth Form | Optimal Band Ratio | | $R^2$ | |
|---|---|---|---|---|---|
| | | Brute Force | Resampling | Brute Force | Resampling |
| chl *a* | Total | 684/674 | 684/674 | 0.57 | $0.57 \pm 0.06$ |
| | Fila/epip | 684/674 | 684/674 | 0.62 | $0.62 \pm 0.05$ |
| | Epilithic | 729/809 | 729/809 | 0.44 | $0.44 \pm 0.06$ |
| Phycocyanin | Total | 753/824 | 753/824 | 0.41 | $0.42 \pm 0.09$ |
| | Epiphytic | 688/655 | 684/674 | 0.33 | $0.59 \pm 0.22$ |
| | Epilithic | 738/804 | 804/738 | 0.50 | $0.51 \pm 0.06$ |

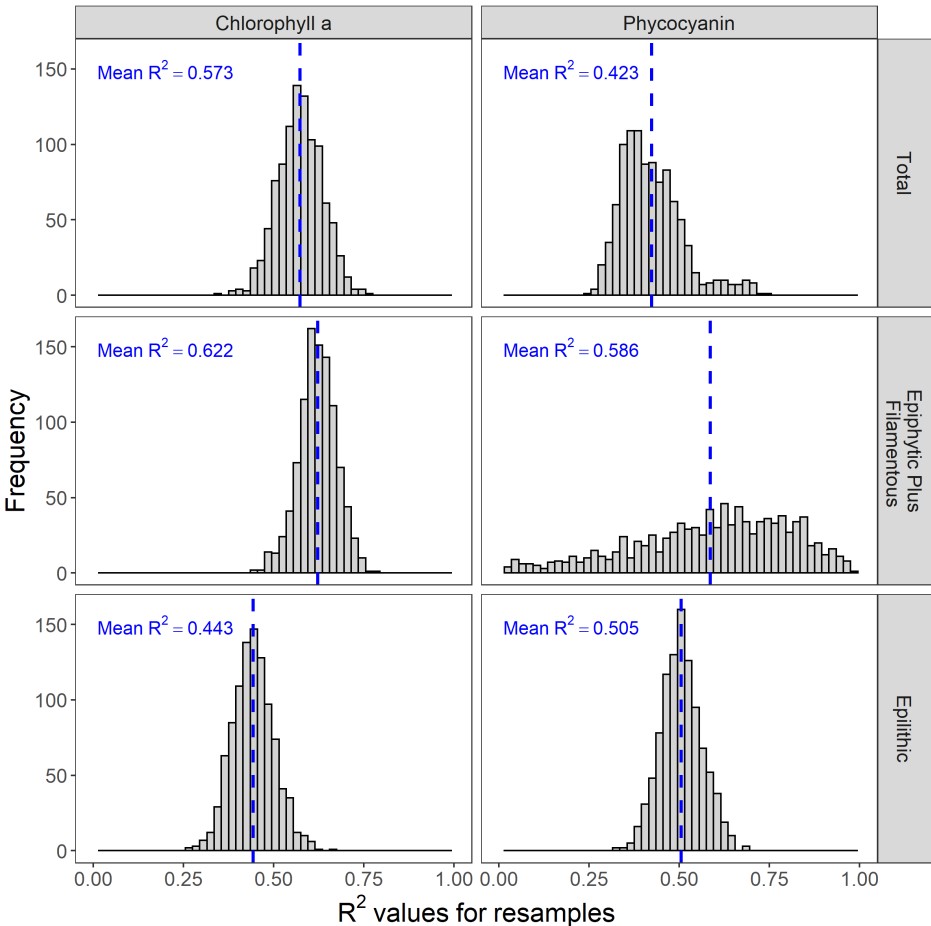

**Figure 7.** Histograms of $R^2$ values generated by running linear regressions on each iteration of pigment subsampling against the optimal band ratios identified for estimating pigment abundance for each growth form.

*3.4. Comparison of Brute-Force Method with Existing Indices*

To provide context for the performance of the presented linear relationships, the derived optimal band ratio for estimating total chl *a* abundance from all field data (n = 33) was compared to several existing band ratio methods for monitoring chl *a*, including the Ha Index [71], Shafique Index [72], and NDVI [73]. All three methods employ the use of band ratios for estimating chl *a* concentration in systems such as tropical lakes using Sentinel 2A imagery [71], large rivers using airborne hyperspectral imagery [72], and general vegetated systems [73]. The selected band ratios were generated using UCFR hyperspectral data, matching the spectral resolution reported in the original work, where NDVI was calculated by integrating across the reported spectral width of Landsat 8 bands [74].

When compared to the optimal band ratio identified by the brute-force method, the selected spectral band ratios performed poorly when tasked with estimating total chl *a* abundance in the UCFR, with a maximum $R^2$ value of 0.13 (Table 5). Surprisingly, although many of the band ratios used similar wavelengths to those selected by the brute-force method, their performance was notably lower, with NDVI having almost no correlation with total chl *a* abundance. The poor performance associated with NDVI is likely attributable to the near-infrared band centered at 865 nm, a spectral region that is strongly absorbed by water. Previous research has shown that the efficacy of NDVI is a function of water depth, with NDVI becoming ineffective for detecting submerged vegetation at depth greater than 30 cm [75].

To explore the effects of spectral resolution on pigment abundance estimations, the spectral channels used to generate NDVI were reduced to 10 nm and 2.1 nm then fit

to pigment abundance. When reduced to 10 nm, the $R^2$ modestly improved to 0.05 ($p$-value = 0.23) while a spectral resolution of 2.1 nm generated similar results ($R^2$ = 0.05, $p$-value = 0.20), suggesting spectral bandwidth did not play a major role in performance in this case. Finally, the brute-force method was set to search for the optimal normalized difference index across the spectral region of 650–850 nm, selecting the same wavelengths used in the simple ratio identified by the brute-force method ($\frac{684 - 674}{684 + 674}$), with nearly identical performance ($R^2$ = 0.57, $p$-value < 0.001).

**Table 5.** Comparison of selected spectral band ratios for estimating total chl *a* abundance in the UCFR from all field sites (n = 33). Listed wavelengths represent band centers for each spectral ratio while regression equations are presented as linear equations in the form of y = mx + b.

| Band Ratio | Center Wavelengths (nm) | Spectral Resolution (nm) | Regression Equation | $R^2$ | *p*-Value |
|---|---|---|---|---|---|
| Ha Index | $\frac{559.8}{664.6}$ | 30, 35 | y = 177.4X − 61.3 | 0.12 | 0.048 |
| Shafique Index | $\frac{705}{675}$ | 5 | y = 154.5X − 47.4 | 0.13 | 0.041 |
| NDVI | $\frac{865 - 655}{865 + 665}$ | 30 | y = -235.5X + 125.8 | 0.04 | 0.25 |
| Brute-Force Method | $\frac{684}{674}$ | 2.1 | y = 2583X − 2448 | 0.57 | <0.001 |
| Brute-Force Method, Normalized Difference | $\frac{684 - 674}{684 + 674}$ | 2.1 | y = 5276X + 135.6 | 0.57 | <0.001 |

## 4. Discussion

The optimal band ratio for estimating total and fila/epip chl *a* standing crops in the combined and Gold Creek data sets is likely explained by a change in the reflectance spectra near a chl *a* absorption region driven by variations in pigment abundance. The denominator selected by the brute-force method is near the spectral trough of this region whereas the numerator is along the red edge, a wavelength pair which is likely sensitive to changes in chl *a* abundance. The shift to shorter wavelengths at Bear Gulch is likely driven by a larger proportion of total BOM derived from epiphytic and epilithic growth, representing later stages of bloom progression. The results presented here suggest that late-stage algae are characterized by a decrease in NIR reflectance, shortening of the red edge, and increased reflectance across visible wavelengths, all of which likely reduce the information contained in the vicinity of the 665 nm chl *a* absorption line. During this progression, spectral information near the visible reflectance peak between approximately 530–600 nm carries more information, leading the brute-force method to this region.

The optimal band ratios selected by the brute-force method show promising generalizability across pigment growth forms assessed in the resampling analysis. The results of this analysis showed elevated maximum mean $R^2$ for the optimal band ratio identified for each growth form, as well as low standard deviation. Though this analysis is still data-limited, these early results show that the optimal band ratios may be a promising tool for assessing pigment abundance in clear and shallow rivers similar to the UCFR.

When compared to existing spectral indices developed for estimating chl *a* concentration, the band ratio selected by the brute-force method showed superior performance to all others, with NDVI performing quite poorly. The poor performance of the existing band ratios may be attributed to their general development for measuring volumetric chl *a* concentration, broader spectral resolution, which tends to smear small spectral features [71,72], or, in the case of NDVI, the use of longer wavelengths that are strongly absorbed by water [73,75]. Overall, the derived band ratios suggest that the spatially averaged spectra (average of all pixels contained in each plot) contain unique and useful information for estimating the abundance of each growth form present.

## 5. Conclusions

A new method using a UAV-based hyperspectral imager for quantitatively characterizing algal blooms in narrow, shallow rivers has been presented. The results from this analysis suggest that spectral band ratios are a promising method for estimating chl *a* and phycocyanin standing crops contained within, near, and on the surface of blooms of the filamentous nuisance algae, *Cladophora glomerata*, growing in the UCFR. The brute-force analysis used to generate spectral band ratios for estimating chl *a* and phycocyanin standing crops from filamentous, epiphytic, and epilithic growth forms often selected wavelengths near known absorption bands of the pigments [69,70]. The absorption bands represent the spectral nature of primary production carried out both by eukaryotic algae and cyanobacteria and reflects their structural and functional character derived from evolutionary selection for light harvesting and carbon fixation. These features are robust and consistent over the time frames applicable to ecological investigations addressing algal abundance; however, the data presented here are limited to the species and conditions present along the UCFR in 2021. Additionally, the scope of this work is limited to the exploration of linear relationships between spectral band ratios and pigment abundance. In cases where the linear fit is not highly correlated (i.e., low $R^2$ values), a nonlinear fit may explain more of the observed variance in pigment abundance, but also raises the risk of over-fitting. Questions still remain about the generalizability of the selected optimal band ratios, meaning that more data must be collected and analyzed, though preliminary analyses show strong agreement across data sets.

The brute-force method converged on two optimal band ratios when estimating fila/epip chl *a* standing crops across data sets, selecting a ratio of 684/674 nm for both the combined and Gold Creek data sets and a ratio of 554/536 nm at the Bear Gulch site. The brute-force analysis appeared to identify other important wavelengths for estimating chl *a* for algae in different phenological states, selecting shorter wavelengths (554, 536 nm) when analyzing *Cladophora* blooms with significant diatom coverage, such as those seen at the Bear Gulch site. The method also illuminates regions sensitive to phycocyanin standing crops, often selecting wavelengths over 700 nm (Table 3). Interestingly, the brute-force analysis selected similar band ratios for epilithic chl *a* and phycocyanin abundance, indicating that these signals may be conflated. That is, the presence of one pigment may be related to the presence (or absence) of the other.

The brute-force analysis outlined here represents the early framework for developing a network of low-cost sensors to estimate pigment standing crops over large stretches of riverine systems affected by algal bloom activity. The adoption of a sensor network based on the proposed method would allow for more frequent data collection, while also providing information on a spatial scale large enough to understand ecological factors such as algal metabolism and nutrient uptake. While current measurement methods are constrained to small spatial scales ($cm^2$–$m^2$), the proposed method allows for much broader spatial coverage (100–1000 m), allowing for greater understanding of the relationship between RAB development and functional attributes related to water quality. Expanding access to RAB detection and monitoring techniques would help maintain water quality standards, especially in areas where standards are based on algal biomass or nutrient standards, or lead to remediation in contaminated waterways.

In future work, a larger data set collected in the UCFR in 2022 will be analyzed and new data will be collected in the Gallatin River in Montana to begin assessing how well the results presented here apply to other times, conditions, and rivers. These additional data also will aid in the design of multispectral imaging systems capable of RAB detection, along with the development of algal classification maps leading to percent cover estimates.

**Supplementary Materials:** The following supporting information can be downloaded at: https://www.mdpi.com/article/10.3390/rs15123148/s1, Figure S1: Analytics from the regression analysis of total phycocyanin standing crops from the combined (all field sites) data set. Figure S2: Analytics from the regression analysis of fila/epip chl *a* standing crops from the combined (all field sites) data set. Figure S3: Analytics from the regression analysis of epiphytic phycocyanin standing crops from the combined (all field sites) data set. Figure S4: Analytics from the regression analysis of epilithic chl *a* standing crops from the combined (all field sites) data set. Figure S5: Analytics from the regression analysis of epilithic phycocyanin standing crops from the combined (all field sites) data set. Figure S6: Analytics from the regression analysis of total chl *a* standing crops from the Bear Gulch data set. Figure S7: Analytics from the regression analysis of total phycocyanin standing crops from the Bear Gulch data set. Figure S8: Analytics from the regression analysis of fila/epip chl *a* standing crops from the Bear Gulch data set. Figure S9: Analytics from the regression analysis of epiphytic phycocyanin standing crops from the Bear Gulch data set. Figure S10: Analytics from the regression analysis of epilithic chl *a* standing crops from the Bear Gulch data set. Figure S11: Analytics from the regression analysis of epilithic phycocyanin standing crops from the Bear Gulch data set. Figure S12: Analytics from the regression analysis of total chl *a* standing crops from the Gold Creek data set. Figure S13: Analytics from the regression analysis of total phycocyanin standing crops from the Gold Creek data set. Figure S14: Analytics from the regression analysis of epiphytic phycocyanin standing crops from the Gold Creek data set. Figure S15: Analytics from the regression analysis of epilithic chl *a* standing crops from the Gold Creek data set.

**Author Contributions:** Conceptualization, R.D.L., H.M.V. and J.A.S.; Data curation, R.D.L., M.A.T. and R.F.-L.; Formal analysis, R.D.L., R.F.-L. and B.P.C.; Funding acquisition, H.M.V. and J.A.S.; Methodology, R.D.L., R.F.-L., H.M.V. and J.A.S.; Project administration, H.M.V. and J.A.S.; Resources, B.P.C., H.M.V. and J.A.S.; Software, R.D.L.; Supervision, B.P.C., H.M.V. and J.A.S.; Validation, B.P.C.; Writing—original draft, R.D.L.; Writing—review & editing, R.F.-L., B.P.C., H.M.V. and J.A.S. All authors have read and agreed to the published version of the manuscript.

**Funding:** This material is based upon work supported in part by the National Science Foundation EPSCoR Cooperative Agreement OIA-1757351. Any opinions, findings, and conclusions or recommendations expressed in this material are those of the author(s) and do not necessarily reflect the views of the National Science Foundation.

**Data Availability Statement:** Publicly available datasets were analyzed in this study. These data can be found here: https://github.com/RileyLogan/UAV-Based-Hyperspectral-Imaging-for-River-Algae-Pigment-Estimation.

**Conflicts of Interest:** The authors declare no conflict of interest.

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
