# Peer review of "UAV-Based Hyperspectral Imaging for River Algae Pigment Estimation"

_remotesensing, doi:10.3390/rs15123148_

Round 1
Reviewer 1 Report
In this paper, the authors used the hyperspectral images captured by UAV to conduct regression analysis of algal pigments in Upper Clark Fork River. The results of chlorophyll a estimation are satisfactory. But there are still some problems in the manuscript that need to be corrected.
1. The most important one is that the authors should addressed that what has been expanded compared with reference 49 published on conference proceedings?
2. The data used in this article are all from the same year. Will data from different years exhibit different characteristics? Then the method used in this paper will do not work?
3. The maximum R2 obtained from linear regression of many parameters is relatively small. Can it indicate that there is a non-linear relationship between the parameters? The use of linear regression methods does not reveal the true relationship between parameters well?
4. No hyperspectral data collected using UAV has been seen in the article yet.
5.In the abstract, the author only described the work done and the results obtained. The author needs to highlight the advantages and characteristics of the proposed method.
6. The introduction section can be narrated in sections, which looks more intuitive and clear.
7. In the line 73 of the manuscript, the authors said "...several established spectral indices have been shown to have decreased performance when generalized across systems with differing water properties. However, no known studies have assessed water quality through estimation of algal standing crops...". It feels more appropriate to undertake rather than turning point-"however", please check carefully.
8. Also in the line 73 of the manuscript--"no known studies have assessed water quality through estimation of algal standing crops...". Was the author's study assessed water quality? I didn't see that part, you just estimated the algae biomass. Please check.
9. In the lines 76-88 of the manuscript, the author introduced the food web of the river and so on. What does this part do? Does it fit here?
10. In the line 89 of the manuscript, the authors said "Current methods for quantifying algal standing crops are constrained to smaller scales than needed to couple structure and function, are hampered by site-specific conditions, and are labor intensive, rendering simple expansion intractable and extrapolation inappropriate." What does "smaller scales", "site-specific conditions" and "labor intensive" mean? The authors should talk more about that here.
11. When "spectral band ratios" first appeared in line 104 of the manuscript, the authors need to explain it.
12. In the line 241 of the manuscript, the authors said "With the flight altitude and field of view, each plot contained an average of approximately 40 pixels, which were averaged to create a single spectrum for each plot." What does this sentence mean? Please explain in detail.
13. What is the Figure S1(line 385)-Figure S15? I didn't find them. Please check. If these pictures are available, please put them somewhere easier to find. If not, add them.
14. In the discussion section, authors need to highlight the advantages of your approach and describe its value, such as the social and economic aspects.
15. The authors need to check more for syntax and formatting issues.
Overall, English writing is acceptable, but there are also some grammar issues.
Reviewer 2 Report
Dear authors,
The paper is well-prepared and well understandable. Thanks to the authors for providing such high-quality materials.
I have a few comments to share:
406 - I noticed that there is no title for the y-axis in Figure 3(c). Is that intentional or should one be added?
536(58) - This is a second abbreviation for NDVI in the text.
Thank you.
Reviewer 3 Report
The authors of this manuscript have done a decent job in the collection of UAV based hyperspectral data. They have used a brute-force regression analysis to estimate chlorophyll-a and phycocyanin. The results achieved from the analysis are interesting and promising. However, there seems to be several areas where the authors would need to provide some more information. The authors need to clearly state the QA-QC procedures for UAV collected hyperspectral data. How the raw data was corrected? did they used any kind of data transformation? how the data was corrected for sun-glint? how outliers were identified and eliminated? How they addressed issues with titling of the UAV, which can affect the downwelling irradiance. The authors have addressed some of those briefly in later sections but that information should be in section 2. In section 3.2 the authors mention visual inspection of heat maps and histograms to determine the optimal band ratios, there are better statistical approaches. Section 3.3 in the uncertainty analysis, the authors need to justify their choice or the resampling or subsampling procedure. The author might consider a bootstrap approach to randomly subsample the data. Next, it would be nice if the authors present the regression equations, beta-coefficients, p-values when comparing estimates from the brute-force with other indices. Figures showing cross-validation, estimated versus measured would also add value to the current manuscript. Finally, it would be nice if the authors could comment why some of the existing indices did not perform well, particularly the NDVI. The current version of the manuscript is not ready for publication, I recommend major revision.
Round 2
Reviewer 1 Report
The manuscript has been improved by the authors. And almost all the comments have been responded well. Supply a hyperspectral image collected using UAV in Figure 3, not the current chart.
Author Response
Thank you for your suggestion, we have updated Figure 3 and believe it adds valuable information and clarification to the reader. However, we have not added a full hyperspectral image to the text as it is impossible because such images contain hundreds of wavelengths at each spatial pixel, which the journal cannot reproduce. Therefore, we created a new version of Figure 3 that shows an RGB-colorized hyperspectral image collected with our airborne UAV system, along with 3 separate examples of the primary spectra that exist within the 3-dimensional hyperspectral image, including bank vegetation, bare river substrate, and algae.
Reviewer 3 Report
Satisfied by the revisions made to the manuscripts.
No more comments.
Author Response
Thank you.